# Research Progress for Targeting Deubiquitinases in Gastric Cancers

**DOI:** 10.3390/cancers14235831

**Published:** 2022-11-26

**Authors:** Tao An, Yanting Lu, Zhaoqi Gong, Yongtao Wang, Chen Su, Guimei Tang, Jingjing Hou

**Affiliations:** 1School of Pharmaceutical Sciences, Qilu University of Technology (Shandong Academy of Sciences), Jinan 250353, China; 2College of Traditional Chinese Medicine, Shandong University of Traditional Chinese Medicine, Jinan 250353, China; 3School of Chemistry and Chemical Engineering, Qilu University of Technology (Shandong Academy of Sciences), Jinan 250353, China; 4Department of Gastrointestinal Surgery, Zhongshan Hospital of Xiamen University, School of Medicine, Xiamen University, Xiamen 361102, China; 5Institute of Gastrointestinal Oncology, School of Medicine, Xiamen University, Xiamen 361005, China

**Keywords:** gastric cancer, deubiquitinases, biomarkers, tumorigenesis, pathogenesis, diagnosis

## Abstract

**Simple Summary:**

The deubiquitinase-mediated cleavage of ubiquitin chains from substrate proteins plays a crucial role in regulating protein degradation, activities, interactions, and localization. The dysregulation of multiple deubiquitinases has been implicated in various human diseases, especially cancer. More importantly, many small molecules targeting oncogenic deubiquitinases have been discovered, some of which have exhibited promising anti-cancer effects and entered clinical trials. Gastric cancer remains one of the most common and fatal malignancies. In this review, we aim to summarize the multifaceted roles of deubiquitinases in gastric tumorigenesis. We also present the upstream regulation of specific deubiquitinases and the research progress of several deubiquitinase-associated small molecules for gastric cancer therapy. Together, this review will improve our understanding of the biological role of deubiquitinases as well as the therapeutic potential of targeting deubiquitinases in gastric cancer.

**Abstract:**

Gastric cancers (GCs) are malignant tumors with a high incidence that threaten global public health. Despite advances in GC diagnosis and treatment, the prognosis remains poor. Therefore, the mechanisms underlying GC progression need to be identified to develop prognostic biomarkers and therapeutic targets. Ubiquitination, a post-translational modification that regulates the stability, activity, localization, and interactions of target proteins, can be reversed by deubiquitinases (DUBs), which can remove ubiquitin monomers or polymers from modified proteins. The dysfunction of DUBs has been closely linked to tumorigenesis in various cancer types, and targeting certain DUBs may provide a potential option for cancer therapy. Multiple DUBs have been demonstrated to function as oncogenes or tumor suppressors in GC. In this review, we summarize the DUBs involved in GC and their associated upstream regulation and downstream mechanisms and present the benefits of targeting DUBs for GC treatment, which could provide new insights for GC diagnosis and therapy.

## 1. Introduction

Gastric cancer (GC) is the fifth most common cancer and the fourth leading cause of cancer-related mortality worldwide. Based on estimates from the GLOBOCAN database, more than 1 million new GC cases and an estimated 769,000 GC deaths occurred worldwide in 2020 [1]. As with other cancer types, GC development is a multistage process involving genetic and epigenetic alterations [2,3]. In addition to host factors, other etiological factors, including a high-salt diet, tobacco use, and infectious agents, such as *Helicobacter pylori* (*H. pylori*) and Epstein–Barr virus (EBV), also play a significant role in the initiation and progression of GC [4]. Despite advances in understanding the mechanisms underlying GC and improved therapies, almost one-third of patients with GC are diagnosed at a late stage, with a 5-year survival rate below 20% [3,5,6]. Therefore, there is an urgent need to develop novel biomarkers and therapeutic targets for treating GC.

Deubiquitinases (DUBs) are isopeptidases that can cleave a single ubiquitin or entire ubiquitin chains from target proteins, thereby counteracting protein ubiquitylation, a post-translational modification fundamental for the regulation of protein stability, activity, subcellular localization, and interactions [7,8]. DUBs are involved in many physiological processes, such as apoptosis, autophagy, and the cell cycle, and pathological processes, such as neurodegenerative diseases and cancers [9,10,11,12,13,14]. Hence, DUBs have attracted attention as therapeutic targets, and DUB inhibitors have been developed, with some now in preclinical development or clinical trials [15]. To date, approximately 100 DUBs have been reported and are classified into seven subfamilies based on their sequence and structural similarities as follows: ubiquitin-specific proteases (USPs), ubiquitin carboxy-terminal hydrolases (UCHs), ovarian tumor proteases (OTUs), Jab1/MPN domain-associated metalloenzymes (JAMMs), Machado–Joseph disease proteases (MJDs), the monocyte chemotactic protein-induced protease family (MINDYs), and Zn-finger and UFSP domain proteins (ZUFSPs) (Figure 1) [16].

During the past decades, many studies have demonstrated that multiple DUBs are implicated in the development of GC. In this review, we first summarize and discuss the DUBs related to GC, with a summary of their mechanisms of action and regulation (Table 1). In addition, we document the potential of pharmacological interventions that target DUBs (Table 2). Taken together, the present review aims to provide a better understanding of the molecular mechanisms underlying GC development associated with DUBs and how they can be targeted for GC treatment.

## 2. USPs and GC

USPs constitute the largest family of human DUBs, with 56 members. As shown in Figure 2, at least 25 USPs are linked to GC.

### 2.1. USP3

Studies have highlighted the role of USP3 in GC progression. USP3 is overexpressed in GC cells and tissues and can serve as a prognostic marker for patients [20,21]. USP3 facilitates GC cell growth and metastasis in vitro and in vivo by modulating cell cycle control and epithelial–mesenchymal transition (EMT)-related molecules [20]. Wu et al. identified differentially expressed proteins in BGC-823 cells stably expressing USP3 [21]. Among them, SUZ12, a scaffolding component of the PRC2 complex, and COL9A3/COL6A5, collagen family members, were deubiquitinated and stabilized by USP3, which accounts for its role in promoting invasion and migration [21,22]. In addition to SUZ12, the core PRC2 complex also comprises the histone methyltransferase EZH2 and the scaffolding component EED. As an epigenetic regulator complex, PRC2 trimethylates lysine 27 on histone H3 tails to modulate its several thousand target genes, including various genes involved in cancer proliferation, migration, and invasion, such as *E-cadherin* and *p14ARF* [114]. However, it is unknown whether these three proteins are involved in the USP3-directed proliferation of GC cells, and future studies should examine their oncogenic roles in GC. 

Additionally, non-coding RNAs, including circular RNAs, long non-coding RNAs (lncRNAs), and microRNAs (miRNAs), play critical roles in modulating GC [115,116,117,118]. Hsa_circ_0017639, a circular RNA, is increased in GC cell lines and promotes proliferation and migration by increasing *USP3* expression by sponging miR-224-5p [23]. In addition, Jin et al. demonstrated that exosomal lncRNA SND1-IT1 secreted from GC cells not only recruited DDX54, a DEAD-box RNA helicase that binds to specific RNAs, to enhance *USP3* mRNA stability, but also bound to miR-1245b-5p to upregulate *USP3* expression, thus leading to SNAIL1 stabilization and inducing the malignant transformation of gastric mucosa cells [24]. Taken together, these studies suggest that targeting USP3 may be a potential treatment for GC.

### 2.2. USP7 and USP11

USP7, also called herpes-virus-associated ubiquitin-specific protease, deubiquitinates many substrate proteins involved in the cell cycle, DNA damage responses, and immune responses [119,120]. Accordingly, the aberrant expression and activity of USP7 have been found in GC. Wang et al. showed that USP7 is highly expressed in GC and increases the expression of programmed cell death protein 1 (PD-L1), a pivotal immune checkpoint molecule that decreases T-cell immune responses [26]. Furthermore, USP7 knockdown or inhibition with its inhibitor Almac4, developed by Gavory et al., conferred GC cell sensitivity to T-cell cytotoxicity, reduced proliferation, and induced cell cycle arrest by stabilizing p53, which interacts with USP7 [26,107,121]. Similarly, the USP7 inhibitor C9, a quinazolin-4(3H)-one derivative, also suppressed GC cell proliferation by upregulating p53 and its downstream target p21 [108]. 

To better understand the USP7 binding network in tumor cells, Anna et al. performed affinity purification coupled with mass spectrometry to identify interactions in GC cells overexpressing USP7. In addition to several reported binding proteins (such as USP11 and TRIP12), this study also identified DHX40 and DDX24, two DEAD/DEAH-box RNA helicases, as novel targets of USP7, providing preliminary evidence that USP7 may regulate RNA metabolism [25]. Furthermore, these interactions were confirmed with nasopharyngeal carcinoma cells. However, the role of these USP7 substrates in GC carcinogenesis were not investigated [25]. Interestingly, a recent study suggested that USP11 might play an oncogenic role in GC by affecting RhoA and Ras-mediated signaling pathways [34]. Moreover, a previous study indicated that *H. pylori* decreases the expression and activity of USP7 in infected GC cells. However, the exact function of *H. pylori* in regulating *USP7* expression remains unclear [122].

In addition, Zhang et al. found that USP7 stabilizes hnRNPA1 in cancer-associated fibroblasts (CAFs) to contribute to the packaging and release of exosomes containing miR-522. Paclitaxel and cisplatin also promoted miR-522 secretion by activating the USP7/hnRNPA1 axis. The lipoxygenase ALOX15 is important in mediating the lipid peroxidation that drives ferroptosis. Further, exosomal miR-522 repressed *ALOX15* expression and lipid peroxide accumulation, inhibited ferroptosis, and ultimately lead to acquired chemoresistance in GC cells [27].

### 2.3. USP9X, USP36, and USP49

USP9X has been shown to function as either an oncogene or tumor suppressor, depending on the type of cancer [123]. Increased USP9X expression is associated with a poor prognosis in patients with GC, suggesting an oncogenic role [28]. Consistent with this evidence, another study showed that silencing *USP9X* represses the migration, invasion, and colony formation ability of GC cells. Moreover, hsa_circ_0008434, a miRNA sponge for miR-6838-5p, increased *USP9X* expression and promoted GC progression [29]. 

Hippo signaling has been implicated in regulating cell growth, metastasis, and chemoresistance in GC [124]. As the core downstream effectors, YAP and TAZ activity is controlled by a conserved kinase cassette. In mammals, once Hippo signaling is activated, MST1/2 kinases phosphorylate and activate LATS1/2 kinases, which further phosphorylate YAP/TAZ for cytoplasmic sequestration or degradation. Inhibiting the Hippo pathway leads to the nuclear translocation of YAP/TAZ, which regulate target gene expression by binding to TEAD coactivators [125,126]. Additionally, several DUBs activate or repress Hippo signaling through different mechanisms [127]. For instance, USP9X promotes breast cancer cell survival and attenuates cell sensitivity to chemotherapy by deubiquitinating and stabilizing YAP [128]. Zhang et al. demonstrated that LINC01433, a lncRNA positively related to GC progression, increased YAP stability by enhancing its interaction with USP9X and decreased YAP phosphorylation by weakening its association with LATS1. Because YAP binds to the LINC01433 promoter and activates its transcription, this positive feedback loop could be a therapeutic target for GC treatment [30].

Two studies have suggested that DUB1 and USP49 also contribute to GC development by regulating the Hippo signaling pathway [58,67]. DUB1 is a short form of USP36, but its role in tumorigenesis remains largely unknown [129,130]. However, it has a role in GC progression. Wang et al. found that DUB1 is capable of interacting with, deubiquitinating, and stabilizing the Hippo signaling effector TAZ [58]. *USP49* silencing decreases cell proliferation, migration, and invasion and enhances GC cell sensitivity to chemotherapy; however, this effect can be reversed by YAP1 overexpression. Because *USP49* is a target gene modulated by YAP1/TEAD4, this result indicates that USP49 and YAP1 form a positive feedback loop to support the malignant progression of GC [67]. 

### 2.4. USP10

USP10 plays a dual role in different human cancer types [131]. Zeng et al. showed that USP10 expression was lower in GC cell lines and clinical samples than in their noncancerous counterparts. More importantly, decreased USP10 expression indicates several highly malignant clinicopathological features and poor survival in patients with GC, suggesting that USP10 may be a prognostic biomarker [31]. Additionally, the calcium-binding protein S100A12 was found to be a GC prognostic marker, and its levels correlated with USP10 [132,133]. Given that USP10 and S100A12 are both located in the cytoplasm, USP10 may regulate the stability of S100A12 via deubiquitylation [132]. Therefore, future studies should investigate the role of USP10 in GC proliferation and metastasis. Moreover, the USP10-mediated stabilization of p53 likely contributes to the sensitivity of GC cells to 3-deazaneplanocin A, a histone methylation inhibitor that depletes EZH2, the enzymatic component of the PRC2 complex that catalyzes H3K27me3 [32]. As p53 turnover is regulated by a variety of DUBs [134], whether these DUBs influence chemotherapy resistance in GC cells warrants further study. While these results indicate that USP10 functions as a tumor suppressor, one study showed that USP10 promoted GC cell migration and invasion by stabilizing replication factor C subunit 2 [33]. Collectively, the function of USP10 in GC remains unclear and requires further studies.

### 2.5. USP13, USP29, and USP37

As a master transcriptional factor that induces EMT, Snail is frequently overexpressed in tumors and drives tumor progression, cell survival, metastasis, and stem cell properties [135]. E3 ligases and DUBs are involved in the phosphorylation-dependent ubiquitination and proteasomal degradation of Snail [135]. For instance, GSK-3β phosphorylates Snail at six Ser sites. Snail phosphorylation at four Ser sites (Ser-107, 111, 115, and 119) causes cytoplasmic localization, whereas phosphorylation at two Ser sites (Ser-96 and 100) promotes ubiquitination and degradation by β-TrCP [136]. USP13, USP29, and USP37 were reported to promote the metastasis of GC by deubiquitinating and stabilizing Snail [35,36,54,59]. In addition, the upstream events of USP29 and USP37 are relevant for Snail regulation. Specifically, USP29, induced by TGFβ, TNFα, and hypoxia, increases the interaction between Snail and phosphatase SCP1, resulting in the dephosphorylation and deubiquitination of Snail, preventing its degradation [54]. USP37, which is transcriptionally activated by PLAG1-like zinc finger 2, stabilizes Snail in a GSK-3β phosphorylation-dependent manner [59]. Overall, these studies demonstrate the complexity of Snail regulation and provide potential therapeutic targets for the treatment of GC.

### 2.6. USP14

USP14 is a proteasome-associated DUB that plays a dual role in regulating protein degradation. Although it cleaves ubiquitin, it also promotes protein degradation by activating the proteasome [137,138]. *USP14* has been reported to be an oncogene in GC [139]. USP14 levels are elevated in GC and may be an independent disease-free survival marker in patients [38,39]. In one study, USP14 was found to promote GC cell proliferation, invasion, and migration by stabilizing the EMT protein vimentin. Furthermore, the authors showed that miR-320a acts as a tumor suppressor by reducing vimentin expression and preventing USP14 from stabilizing this protein [37]. However, Fu et al. showed that *USP14* depletion did not lead to cell death but sensitized GC cells to cisplatin-induced apoptosis [38]. As one chemical modification commonly found in eukaryotic mRNAs, N^6^-methyladenosine (m^6^A) plays a crucial role in modulating mRNA processing; studies have indicated that dysregulated m^6^A regulators are involved in cancer progression [140]. The m^6^A reader YTH N^6^-methyladenosine RNA binding protein 1 (YTHDF1) promotes gastric carcinogenesis by modulating the translation of FDZ7, a key Wnt receptor [141]. YTHDF1 also increased USP14 levels in an m^6^A-dependent manner, and USP14 overexpression reversed the tumor-suppressive effects elicited by *YTHDF1* silencing [39]. Additionally, the USP14 inhibitor IU1 restricted GC cell viability and metastasis triggered by YTHDF1 overexpression [39,109]. Together, these findings provide novel insights into the USP14-related mechanisms underlying malignancy in GC. 

### 2.7. USP15

The role of USP15 in GC is less defined and contradictory. Zheng et al. suggested that the ectopic expression of USP15 could suppress GC cell growth, migration, and invasion through the deubiquitination of IκBα by the COP9 signalosome (CSN) complex, thereby hindering NF-κB activity [40,142]. In contrast, *USP15* knockdown attenuated the activity of Wnt/β-catenin signaling and inhibited GC progression both in vitro and in vivo, but the mechanism through which USP15 regulates this pathway remains unclear [41]. As reviewed by Das et al., USP15 can either activate or suppress NF-κB and Wnt/β-catenin signaling depending on its effects on different proteins in various cellular contexts [143]. Furthermore, Huangfu et al. suggested that USP15 may assist in GC development by acting as a target that is regulated by the LINC00205/miR-26a axis [42]. In short, the effects of USP15 on specific signaling pathways, such as NF-κB and Wnt/β-catenin, and on tumorigenesis and metastasis in GC require further study.

### 2.8. USP22

USP22 has been identified as producing cancer-stem-cell-like qualities, including aggressive growth, metastasis, and therapy resistance [144]. Unlike other DUBs, the USP22 zinc-finger ubiquitin-binding domain does not directly bind to ubiquitin and instead is recruited and activated by the SAGA complex, forming a subcomplex (DUB module) with other proteins [145,146]. By regulating a broad range of histone and non-histone protein substrates, USP22 is associated with oncogenesis in GC [145]. Several studies have suggested that USP22 expression is upregulated in GC tissues and cells [45,46,47,49,50,51,52]. In gastric tumors, USP22 expression was found to be positively correlated with the expression of the three well-known oncoproteins BMI1, c-Myc, and HSP90, which better predicted GC progression and prognosis than other methods [45,46,49]. Notably, USP22 stabilizes BMI1 and c-Myc, but not HSP90 [47,49,50]. USP22 maintained GC cell stemness by stabilizing BMI1 and promoted proliferation and metastasis by activating the FOXO1 and YAP signaling pathway [47,50]. However, these studies did not provide evidence that USP22 can stabilize BMI1 or c-Myc through deubiquitination, but this role was later identified in glioma and breast cancer cells [147,148]. As a guanine nucleotide exchange factor, son of sevenless 1 (SOS1) catalyzes GTP-bound RAS formation, whose activation elicits oncogenic signaling pathways [149]. Lim et al. proposed that USP22 upregulates SOS1 expression in GC, thus leading to the activation of RAS/ERK and RAS/PI3K/AKT pathways [52]. USP22 can also be modulated by the POU2F1/miR-4490 axis to further increase GC proliferation and metastasis [51].

To target these USP22-regulated stemness properties, Yang et al. constructed *USP22* siRNA-loaded nanoliposomes modified with anti-CD44 (USP22-NLs-CD44), which targets a stem-cell-associated marker. USP22-NLs-CD44 impaired tumorsphere formation and reduced the percentage of CD44^+^ cells in two human GC cell lines, MKN45 and NCI-N87, presenting an approach to eliminating GC stem cells [48]. 

Although the aforementioned studies indicate that *USP22* is a GC oncogene, a meta-analysis noted contradictions between USP22 expression, tumor size, differentiation state, tumor stage, and clinical outcomes in patients with gastric tumors. However, these inconsistencies may be due to the small sample size, suggesting that the role of USP22 should be analyzed in larger randomized controlled clinical trials [150]. Moreover, there are discrepancies in USP22′s effects on malignant behaviors. Ma et al. reported that USP22 only affects cell proliferation, whereas other authors have shown that USP22 affects GC cell proliferation, cell migration, and apoptosis [47,50,51,52]. Such controversial conclusions may be explained by differences in the cell context, including variations in USP22 levels among cell lines. Taken together, the role and downstream effectors of USP22 in GC warrant further investigation.

### 2.9. USP28

USP28 has been validated as an oncoprotein and therapeutic target in various cancer types, and a list of oncogenic substrates of USP28 have been reported, such as JUN, NOTCH1, CCNE, and LSD1 [151,152]. For instance, USP28 was identified as a DUB of LSD1 and conferred stem-cell-like characteristics in breast cancer [153]. Likewise, Zhao et al. found that USP28 was highly expressed in GC and was conducive to proliferation and metastasis, mainly due to its ability to increase LSD1 levels [53]. The same group subsequently designed and synthesized new [1,2,3] triazolo [4,5-d] pyrimidine derivatives as potent USP28 inhibitors. Among them, compound 19 potently inhibited USP28 activity with an IC_50_ of 1.10 ± 0.02 μM and induced its degradation at higher doses, thereby exhibiting cytotoxic effects against GC cells [110]. In addition to LSD1, compound 19 also promotes c-Myc degradation, another substrate of USP28 [110]. Hu et al. showed that two GC cell lines, MKN-45 and SGC-7901, were sensitive to lanatoside C, an FDA-approved cardiac glycoside derived from *Digitalis lanata*. They also showed that lanatoside C partially promoted c-Myc degradation by reducing its interactions with USP28 [111]. These studies suggest that c-Myc is a potential downstream effector responsible for the oncogenic role of USP28 in GC.

### 2.10. USP32

USP32 was reported to be one protein that contributed to the chimeric *Tre2* (*USP6*) oncogene [154]. The pro-cancer effects of USP32 have been observed in breast cancer and glioblastoma [155,156]. Additionally, USP32 promotes GC cell growth, metastasis, and chemoresistance by upregulating SMAD2, an important protein in the TGF-β signaling pathway [55]. However, it is unclear whether USP32 regulates the ubiquitination level of SMAD2.

### 2.11. USP33

USP33 expression is reduced in GC tissues and cell lines, which correlates with poor patient survival [56,57]. Additionally, USP33 overexpression suppresses the proliferation, migration, and invasion of gastric adenocarcinoma cells [56]. In addition, Xia et al. demonstrated that USP33 deubiquitinates and stabilizes ROBO1, which is required for the SLIT2-mediated inhibition of EMT and GC cell metastasis [57].

### 2.12. USP39

Although USP39 was initially identified as a DUB without ubiquitin hydrolysis activity, other studies have suggested that it plays a role in RNA splicing and oncogenesis in multiple malignant tumors [157,158]. Intriguingly, USP39 stabilizes SP1, ZEB1, and FOXM1 through its deubiquitination activity, thus supporting the progression of hepatocellular carcinoma and breast cancer [159,160,161]. Silencing *USP39* expression hinders MGC-803 GC cell proliferation and induces G2/M arrest and PARP cleavage [60]. In line with these results, *USP39* knockdown also decreased the proliferation of MGC-803 and HGC-27 cells. Moreover, miR-133a can suppress *USP39* expression by directly targeting its 3′ UTR, thereby acting as a tumor suppressor in GC [61]. However, the mechanisms by which *USP39* depletion inhibits GC proliferation require further investigation.

### 2.13. USP44

Aneuploidy is a common characteristic of tumor cells [162]. Several studies have demonstrated that USP44 dysregulation results in aneuploidy, which has also been observed in GC tissues [63,163]. Furthermore, the combination of elevated USP44 expression and DNA aneuploidy provides useful prognostic information [63]. In addition, *USP44* is regulated by the circFOXO3/miR-143-3p axis in GC, in which circFOXO3 functions as a miR-143-3p sponge to promote USP44-directed malignancy in gastric carcinoma [64].

### 2.14. USP47

*USP47* is inhibited by miR-204-5p, which represses GC cell proliferation [65]. Interaction between USP47 and β-TrCP, a subunit of the SCF β-TrCP E3 ubiquitin ligase complex, is involved in a number of signaling pathways [164,165,166]. Naghavi et al. found that USP47 stabilizes β-TrCP-modulated NF-κB activity by promoting RelA phosphorylation. Their findings also showed that *USP47* knockdown sensitized NCI-N87 cells to camptothecin- or etoposide-induced apoptosis [66]. These data indicate that USP47 may be involved in NF-κB-dependent chemoresistance in some cellular contexts.

### 2.15. CYLD

CYLD was initially described in cylindromatosis and is currently considered a tumor suppressor that negatively regulates multiple signaling pathways, such as NF-κB, AKT, and Wnt [167,168,169]. Ghadami et al. showed that CYLD expression was decreased in GC tissues owing to the hypermethylation of its promoter. They also revealed a direct correlation between *H. pylori*, EBV, and CMV infections and *CYLD* hypermethylation and downregulated protein expression, suggesting that infectious-agent-induced *CYLD* hypermethylation may be a significant mechanism of GC development [73]. Several studies have shown that CYLD can also be regulated by non-coding RNAs and serve as a downstream effector by modulating the progression and chemoresistance of GC. For example, miR-130b, miR-425-5p, miR-454, and exosomal miR-588 from M2-polarized macrophages negatively regulated *CYLD* expression by targeting its 3′-UTR, thus promoting GC cell proliferation, migration, invasion, and treatment resistance [70,72,75,76]. In addition, the activation of NF-κB by miR-362, miR-500, and miR-20a plays a significant role in the survival and cisplatin resistance of GC cells [68,69,71]. Notably, miR-500 also activates NF-κB signaling by directly repressing two other negative regulators of NF-κB, *OTUD7B*, and *TAX1BP1* [69]. miR-505 also targets the 3′-UTR of *CYLD* mRNA. However, miR-505 can be sponged by CRAL and increase CYLD levels, which decreases AKT signaling and increases treatment susceptibility in GC cells [74]. Furthermore, the ALKBH5/ZNF333/CYLD axis participates in the development of gastric intestinal metaplasia, a precursor of GC. In bile-acid-induced gastric intestinal metaplasia, ZNF333 activates NF-κB signaling by repressing CYLD expression [77]. Taken together, these studies suggest that targeting proteins and miRNA that reduce CYLD levels may be a useful therapeutic strategy for GC.

### 2.16. Other USPs Involved in GC

In addition to the USPs mentioned previously herein, there are several other GC-associated USPs, including USP1, USP2, USP20, USP21, and USP42. USP1 levels were increased in GC cell lines and clinical samples, and its overexpression confers poor patient survival rates. USP1 knockdown inhibits their proliferation, migration, and invasion [17,18]. Mechanistically, USP1 promotes metastasis by stabilizing the inhibitor of DNA binding-2 protein [18]. USP2 was recently reported to harbor oncogenic properties by promoting E2F4-mediated cytoprotective autophagy and zinc homeostasis. By blocking the USP2-E2F4 interaction, emetine inhibited autophagy and GC aggressiveness, suggesting the therapeutic potential of targeting the USP2-E2F4 axis [19].

USP20 promotes the tumorigenesis of several cancer types, such as breast and cervical cancer; however, it exhibits inhibitory effects on GC [16]. Past reports have indicated that USP20 overexpression inhibited cell proliferation and delayed the cell cycle transition from the G1 to S phase by stabilizing Claspin [43,170]. Additionally, USP21 elevates MAPK1 expression through the transcription factor GATA3, thereby contributing to tumor growth and stemness in GC [44]. Hou et al. showed that higher USP42 expression was associated with poor prognosis in patients with GC. In vitro and in vivo studies showed that *USP42* knockdown inhibited GC cell survival and invasion, suggesting that USP42 may be a therapeutic target [62].

## 3. UCHs and GC

All members of this family have been reported to be associated with GC (Figure 3). UCHL1, also called PGP9.5, controls intracellular ubiquitin levels and is related to tumorigenesis in various cancer types [171]. However, UCHL1 has a paradoxical role in GC. *UCHL1* promoter hypermethylation is a common event in several types of primary digestive tumors, including GC [78,79,80,81,82]. Moreover, *UCHL1* hypermethylation is more frequent in advanced-stage GC [78,81]. In addition, this hypermethylation is associated with poor prognosis [82]. Furthermore, galangin cytotoxicity in SNU-484 cells may occur through increasing UCHL1 levels [112]. These studies indicate that UCHL1 may serve as a tumor suppressor and a diagnostic marker for GC. However, two other studies reported higher *UCHL1* expression in liver metastases from GC and gastric cardiac adenocarcinoma, likely because *UCHL1* overexpression increases the proliferation, migration, and invasion capabilities of GC cells [83,84]. Lastly, the pro-cancer effect of UCHL1 is mechanistically correlated with activated Akt and ERK1/2 pathways [83]. Thus, the role of UCHL1 in GC requires further investigation.

Similar to USP14 and PSMD14, UCHL5 is also a proteasome-associated DUB. Higher UCHL5 expression is a predictor of increased survival in a subgroup of patients with early-stage GC [86]. However, another study suggested that UCHL5 stabilizing NFRKB, a chromatin-remodeling protein, may promote GC cell proliferation and metastasis [87]. Therefore, the biological function of UCHL5 requires more investigation.

One recent study reported that UCHL3 promotes GC cell migration and invasion by upregulating IGF2 [85]. In addition, Yan et al. found that BAP1 levels are decreased in GC, which was linked to advanced tumor features and unfavorable survival, suggesting its role as a tumor suppressor [88]. Moreover, another study suggested that BAP1 upregulation is essential for ferroptosis induced by 3,3′-diindolylmethane in BGC-823 cells, suggesting that BAP1-induced ferroptosis could be one of the potential mechanisms by which it suppresses GC progression [113].

## 4. OTUs and GC

Three members of this family, including OTUB1, OTUB2, and A20, have been linked to GC (Figure 4). Weng et al. found that patients with GC with high OTUB1 expression were associated with several advanced clinical features, such as invasion depth, lymph node status, and nerve invasion, and these patients had lower disease-specific survival rates. OTUB1 was found to be active in GC cell invasion and migration [89], but the underlying mechanism is unknown. Similar to OTUB1, OTUB2 was also found to be overexpressed in GC tissues and cell lines and predicted a poor prognosis [90,91]. Silencing OTUB2 inhibited GC cell growth, metastasis, and sphere formation. Mechanistically, OTUB2 acts as a potential driver oncogene in GC by deubiquitinating and stabilizing the demethylase KDM1A, and epithelial keratin KRT80 [90,91]. As previously reported, KDM1A and KRT80 contributed to GC progression by regulating KLF2 expression and the PI3K/Akt pathway, respectively [172,173].

A20, also known as TNFAIP3, is a ubiquitin-editing enzyme with both DUB and E3 ligase activities [174]. A20 expression is increased in GC tissues and cell lines, which is associated with poor clinical outcomes [92,93]. In vitro studies have suggested that *A20* downregulation suppresses the proliferation, migration, and invasion of MGC-803 GC cells [93]. In addition, *A20* was found to be the target of miR-200a in GC cells (MGC-803 and SGC7901) resistant to TRAIL-induced apoptosis. Overexpressing miR-200a or depleting *A20* enhanced apoptosis by reducing RIP1 polyubiquitination and promoting caspase-8 cleavage [94]. These results were consistent with a previous study showing that A20 mediates the polyubiquitination of RIP1, which in turn binds to and inhibits caspase-8 activation [175]. Later studies suggested that RIP1 is involved in multiple cellular signaling pathways and processes, such as NF-κB activation and apoptosis, and promotes GC growth and invasion [176]. Moreover, the DUB activity of A20 counteracted the ubiquitination of procaspase-8, thereby restricting caspase-8 activity and apoptosis [96]. Consistently, NF-κB activation induced *A20* upregulation during *H. pylori* infection, mainly because USP48 stabilizes nuclear RelA and promotes its transcriptional activity [95,96,97,98]. Together, USP48 and A20 promoted GC cell survival during *H. pylori* infection and suggested an oncogenic role for A20. Conversely, one study reported that *H. pylori* infection increased miR-29a-3p, which promoted the migration of gastric epithelial cells by reducing *A20* expression, indicating that *A20* silencing may induce EMT to promote GC progression [99]. Thus, the role of A20 in GC might be context-dependent and requires further investigation.

## 5. JAMMs and GC

Three members of this family, PSMD14, CSN5, and BRCC3, have been found to be associated with GC (Figure 5). PSMD14, also known as Rpn11 and POH1, is a subunit of the proteasomal 19S regulatory particle, which functions as a DUB [177]. PSMD14 overexpression has been reported to be tumorigenic and promote cancer progression through multiple mechanisms [177,178,179,180,181], such as stabilizing the alternative splicing factor PTBP1 to promote GC tumorigenesis [100].

CSN5, the catalytic subunit of the COP9 signalosome, also called COPS5 or JAB1, may play a role in GC [182]. CSN5 overexpression contributes to GC by modulating the stability or expression of several tumorigenic proteins. Silencing *CSN5* suppresses GC cell growth and induces apoptosis by regulating *P53* and *BAX* expression [102]. Additionally, CSN5 induces non-ubiquitin proteasomal degradation of the tumor suppressor p14ARF [103]. CSN5 also facilitates the nuclear export and degradation of the tumor suppressor RUNX3 [101]. Previous studies suggested that TNFα and CCL5 increase CSN5, stabilize PD-L1, and facilitate the immune escape of breast cancer cells and colorectal cancer cells [183,184]. CSN5 also stabilizes PD-L1 in GC cells. Furthermore, CSN5 activity in GC cells is regulated by the DAPK1/IKKβ axis [104]. Collectively, these studies indicate that CSN5 could be a novel therapeutic target in GC.

Lastly, Hu et al. reported that BRCC3 is upregulated in GC and is regulated by the lncRNA TMPO-AS1/miR-126-5p axis. TMPO-AS1 sponges miR-126-5p to upregulate *BRCC3* expression, thereby activating the PI3K/Akt/mTOR pathway, which leads to malignancy [105].

## 6. MJDs and GC

In the MJD family, only Ataxin-3 has been reported to be associated with GC tumorigenesis. Ataxin-3 levels were found to be decreased in GC tissues and cells, which correlated with clinicopathological characteristics, including tumor size, Lauren classification, histologic differentiation, and p53 mutation status [106]. However, the molecular mechanisms underlying this process have not yet been elucidated.

## 7. Conclusions and Perspectives

In the past few years, DUBs and tumorigenesis have been linked. Here, we summarize the regulatory roles of DUBs in the occurrence and development of GC. As shown in Figure 2, Figure 3, Figure 4 and Figure 5, most DUBs promote GC progression, whereas several DUBs play an inhibitory role or exert context-dependent effects. Moreover, the regulatory mechanisms of DUBs are complicated and involve multiple targets and signaling pathways, yet some regulatory mechanisms have not been discovered (Table 1). Although the USP subfamily has received the most attention in GC research, other subfamilies, namely, OTUs, UCHs, JAMMs, and MJDs, have received attention as well. However, there have been no reports on MINDYs and ZUFSPs to date. Therefore, we speculate that USPs may be the most promising biomarkers for GC diagnosis and treatment, and we believe that the function of other DUB subfamilies requires further in-depth exploration.

In addition, DUB regulation is complex. Although the regulatory mechanisms of some GC-related DUBs remain to be discovered, non-coding RNAs have been reported to modulate the expression of DUBs, such as USP3 and CYLD, in GC (Table 1). Intriguingly, three studies discovered that non-coding RNAs are also key mediators of interactions between GC cells and other cell types, including gastric mucosa cells, CAFs, and M2-polarized macrophages, and that USP3, USP7, and CYLD are linked to this process [24,27,76]. Furthermore, several infectious agents, cytokines, and antitumor agents also affect the expression of several DUBs, such as USP3, USP7, USP10, USP29, CYLD, and A20 [21,27,32,54,73,97,98,122] (Table 1), which implicates unknown pro-cancer effects under certain circumstances.

Finally, research on the therapeutic potential of DUB inhibitors for GC therapy is limited despite the efforts made in their development. Some inhibitors show promising therapeutic effects in various cancer types [15,185]. For instance, pharmacological inhibition of USP28, USP1, and USP7 with their inhibitors efficiently hindered in vitro and in vivo tumor growth or metastasis in squamous cell carcinoma, breast cancer, and colon cancer, respectively [186,187,188]. However, only a few studies have demonstrated the antitumor effects of USP7, USP14, and USP28 inhibitors on GC cells. Moreover, compounds that upregulate DUB tumor suppressors also exhibited activity against GC cells, providing another direction for drug development (Table 2). In conclusion, the investigation of DUBs in the pathogenesis and treatment of GC requires more work, which may provide clues for GC treatment in the future.

## Figures and Tables

**Figure 1 cancers-14-05831-f001:**
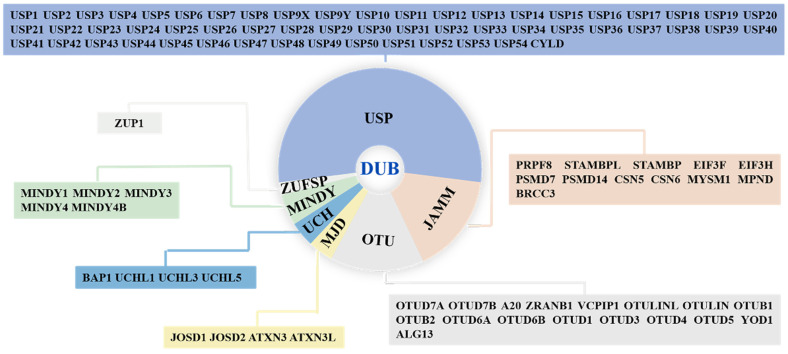
The classification and members of DUB family. DUBs are divided into seven subfamilies: ubiquitin-specific proteases (USPs), ubiquitin carboxy-terminal hydrolases (UCHs), ovarian tumor proteases (OTUs), Jab1/MPN domain-associated metalloenzymes (JAMMs), Machado–Joseph disease proteases (MJDs), monocyte chemotactic protein-induced protease family (MINDYs), and Zn-finger and UFSP domain proteins (ZUFSPs).

**Figure 2 cancers-14-05831-f002:**
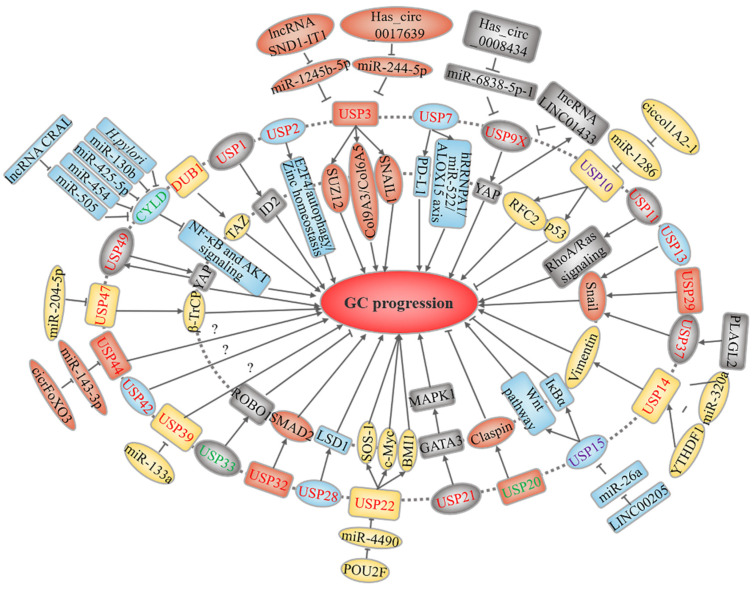
USP-related upstream regulation and downstream mechanisms in GC. USPs marked in red and green represent oncoproteins and tumor suppressors, respectively, while purple-labeled USPs play a dual role. The text in the outer and inner circles of the USPs describes the upstream regulatory events and downstream substrates or signaling pathways, respectively.

**Figure 3 cancers-14-05831-f003:**
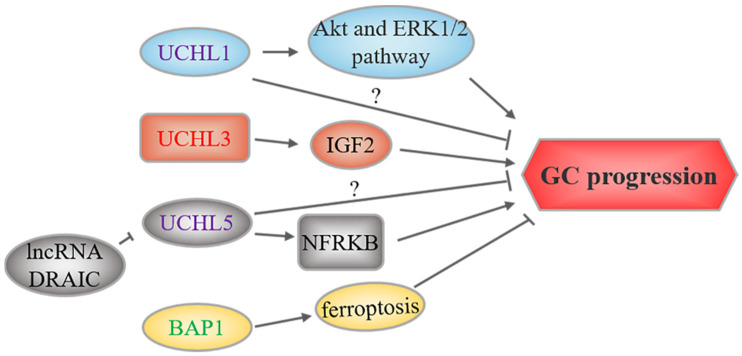
UCH-related upstream regulation and downstream mechanisms in GC. UCHs marked in red and green represent oncoproteins and tumor suppressors, respectively, while purple-labeled UCHs play a dual role. The text to the left and right of the UCHs corresponds to the upstream regulatory events and downstream substrates or signaling pathways, respectively.

**Figure 4 cancers-14-05831-f004:**
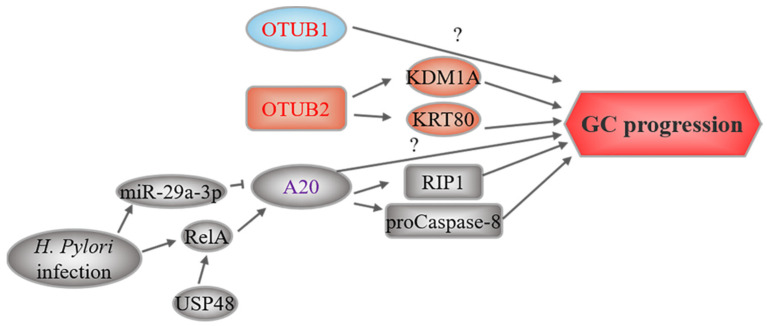
OTU-related upstream regulation and downstream mechanisms in GC. OTUs marked in red represent oncoproteins, while purple-labeled OTUs play a dual role. The text to the left and right of the OTUs corresponds to the upstream regulatory events and downstream substrates, respectively.

**Figure 5 cancers-14-05831-f005:**
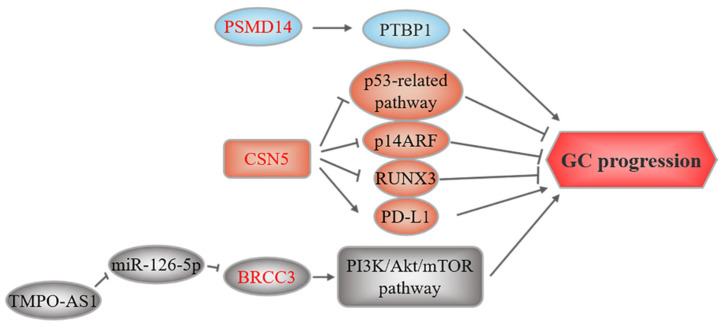
JAMM-related upstream regulation and downstream mechanisms in GC. These three members of JAMMs all play an oncogenic role. The text to the left and right of the JAMMs corresponds to the upstream regulatory events and downstream substrates, respectively.

**Table 1 cancers-14-05831-t001:** The list of DUBs involved in GC.

DUBs	Upstream Regulatory Events	Substrate(s)	Biological Effects	Ref.
USPs				
USP1	None reported	None reported	*USP1* silencing hindered the proliferation, migration, and invasion of GC cells.	[17]
	None reported	ID2	USP1 promoted GC metastasis by stabilizing ID2.	[18]
USP2	None reported	E2F4	USP2 promoted GC progression by facilitating E2F4/autophagy/zinc homeostasis axis.	[19]
USP3	None reported	None reported	USP3 promoted GC cell proliferation and spreading by regulating cell-cycle-control- and EMT-related molecules.	[20]
	TGF-β upregulates *USP3* expression	SUZ12	USP3 promoted GC metastasis by stabilizing SUZ12.	[21]
	None reported	COL9A3/COL6A5	USP3 promoted GC cell migration, invasion, and EMT via binding to and deubiquitinating COL9A3 and COL6A5.	[22]
	Hsa_circ_0017639 upregulates *USP3* expression by sponging miR-224-5p	None reported	Hsa_circ_0017639 participates in GC progression by regulating the miR-224-5p/USP3 axis.	[23]
	LncRNA SND1-IT1 functions as a ceRNA to upregulate *USP3* expression via absorbing miR-1245b-5p and simultaneously recruiting DDX54 to enhance USP3 mRNA stability	SNAIL1	Exosomal SND1-IT1 from GC cells upregulated USP3 expression, thus mediating SNAIL1 stabilization and accelerating the migration and invasion of gastric mucosa cells.	[24]
USP7	None reported	USP11, PPM1G, DHX40, DDX24 and TRIP12	Unclear	[25]
	None reported	PD-L1	USP7 served as an upstream DUB of PD-L1; USP7 abrogation hindered GC cell growth by downregulating PD-L1-mediated immunosuppression and enhanced cell cycle arrest simultaneously by stabilizing p53.	[26]
	Cisplatin and paclitaxel promoted *USP7* expression in CAFs	hnRNPA1	Cisplatin and paclitaxel promoted the secretion of miR-522 from CAFs by activating USP7/hnRNPA1 axis, resulting in ferroptosis suppression, and acquired chemoresistance by inhibiting *ALOX15* expression and lipid-ROS accumulation in GC cells.	[27]
USP9X	None reported	None reported	USP9X was overexpressed and predicted poorer survival in GC.	[28]
	Hsa_circ_0008434 enhances *USP9X* expression by sponging miR-6838-5p	None reported	Hsa_circ_0008434 promoted GC proliferation, invasion, and migration by regulating miR-6838-5p/USP9X axis.	[29]
	None reported	YAP	The LINC01433-YAP feedback loop promoted GC cell proliferation, migration, invasion, and chemotherapy resistance. LINC01433 increased the stability but decreased the phosphorylation of YAP by enhancing its interaction with USP9X and attenuating its interaction with LATS1, respectively.	[30]
USP10	None reported	None reported	USP10 was an independent prognostic marker for patients with GC.	[31]
	3-Deazaneplanocin A treatment upregulates *USP10* expression by reducing EZH2 binding on its promoter	p53	Stabilization of p53 by USP10 seemed to be correlated with the sensitivity of GC cells to 3-Deazaneplanocin A.	[32]
	CircCOL1A2 upregulates *USP10* expression by sponging miR-1286	RFC2	CircCOL1A2 sponges miR-1286 to promote GC cell migration and invasion by increasing USP10 level to stabilize RFC2.	[33]
USP11	None reported	None reported	USP11 overexpression promoted proliferation and migration and alleviated paclitaxel’s toxicity in GC cells by inhibiting RhoA and Ras signaling.	[34]
USP13	None reported	None reported	High expression of USP13 predicted poor prognosis in GC.	[35]
	None reported	Snail	USP13 promoted the EMT and metastasis of GC cells by stabilizing Snail.	[36]
USP14	miR-320a inhibits *USP14* expression by targeting its 3′-UTR	Vimentin	USP14-mediated deubiquitination of vimentin enhanced the aggressiveness of GC cells, and miR-320a served as a tumor suppressor by inhibiting both *USP14* and *vimentin*.	[37]
	None reported	None reported	*USP14* silencing sensitized GC cells to cisplatin by impeding Akt/ERK signaling pathways.	[38]
	YTHDF1 enhanced USP14 protein translation in a m^6^A-dependent manner	None reported	YTHDF1 promoted GC progression and metastasis by promoting USP14 protein translation in an m^6^A-dependent manner.	[39]
USP15	None reported	None reported	USP15 overexpression inhibited GC cell proliferation, migration, and invasion.	[40]
	None reported	None reported	USP15 promoted GC progression through the Wnt/β-catenin signaling pathway.	[41]
	*USP15* is potential regulated by LINC00205/miR-26a axis	None reported	Unclear	[42]
USP20	None reported	None reported	USP20 inhibited GC cell growth and G1/S cell cycle transition by regulating Claspin.	[43]
USP21	None reported	GATA3	USP21 upregulated MAPK1 expression by stabilizing GATA3 to promote GC cell growth and stemness.	[44]
USP22	None reported	None reported	Coordinate expression of USP22 and BMI1 correlated with GC progression and treatment failure.	[45]
	None reported	None reported	High expression of USP22 correlated with GC progression and has synergistic effects with c-Myc.	[46]
	None reported	BMI1	USP22 contributed to gastric CSC stemness maintenance and GC progression by stabilizing BMI1.	[47]
	None reported	None reported	*USP22* siRNA-loaded nanoliposomes decorated with CD44 antibodies selectively target and eliminate CD44^+^ GC stem cells.	[48]
	None reported	None reported	Positive co-expression of USP22 and HSP90 might be more effective in predicting prognosis of GC.	[49]
	None reported	None reported	USP22 promotes GC progression and metastasis through c-Myc/NAMPT/SIRT1-dependent FOXO1 and YAP signaling.	[50]
	POU2F1 upregulates *USP22* expression by suppressing the expression of miR-4490	None reported	POU2F1-miR-4490-USP22 signaling axis plays a significant role in GC development and progression.	[51]
	None reported	None reported	USP22 overexpression in GC induces the upregulation of SOS1 and activation of the RAS/ERK and PI3K/AKT pathways.	[52]
USP28	None reported	None reported	USP28 promoted cell proliferation and metastasis of GC cells by regulating LSD1.	[53]
USP29	TGFβ, TNFα, and hypoxia induced the transcription of *USP29*	Snail	USP29 cooperated with SCP1 to prevent Snail degradation and further promoted GC cell metastasis.	[54]
USP32	None reported	SMAD2	USP32 is involved in GC development and chemoresistance through the upregulation of SMAD2.	[55]
USP33	None reported	None reported	USP33 overexpression inhibited GC cell proliferation, migration, and invasion.	[56]
	None reported	ROBO1	USP33 contributed to SLIT2-ROBO1 axis in inhibiting GC cell migration and EMT process.	[57]
DUB1	None reported	TAZ	DUB1 stabilized TAZ protein and promoted GC progression via the Hippo/TAZ axis.	[58]
USP37	PLAGL2 activated *USP37* transcription	Snail	USP37, which is transcriptionally activated by PLAGL2, deubiquitinates and stabilizes Snail1 to promote the proliferation, EMT, and metastasis of GC cells.	[59]
USP39	None reported	None reported	*USP39* silencing inhibited the growth of GC cells via PARP activation.	[60]
	miR-133a inhibited *USP39* expression by targeting its 3′-UTR	None reported	miR-133a suppressed GC proliferation by regulating *USP39*.	[61]
USP42	None reported	None reported	*USP42* depression inhibited GC cell proliferation and invasion.	[62]
USP44	None reported	None reported	USP44 overexpression resulted in DNA aneuploidy.	[63]
	CircFOXO3 upregulates *USP44* expression by sponging miR-143-3p	None reported	CircFOXO3 promoted GC cells proliferation and migration by increasing *USP44* expression through targeting miR-143-3p.	[64]
USP47	miR-204-5p inhibited *USP47* expression via binding its 3′-UTR	None reported	MiR-204-5p functioned as a tumor suppressor in GC by suppressing *USP47* and *RAB22A*.	[65]
	None reported	None reported	USP47 regulates NF-κB activity by stabilizing β-TrCP and contributes to chemoresistance of GC cells.	[66]
USP49	*USP49* was transcriptionally activated by the YAP1/TEAD4 complex	YAP1	USP49 and YAP1 form a positive feedback loop to promote malignant progression of GC.	[67]
CYLD	miR-362 inhibited *CYLD* expression via targeting its 3′-UTR	None reported	Upregulated miR-362 promoted GC cell proliferation and cisplatin resistance by repressing *CYLD* and activating NF-κB signaling.	[68]
	miR-500 inhibited *CYLD* expression via targeting its 3′-UTR	None reported	Upregulated miR-500 promoted GC cell proliferation and cisplatin resistance by repressing *CYLD*, *OTUD7B*, and *TAX1BP1* and activating NF-κB signaling.	[69]
	miR-130b inhibited *CYLD* expression via targeting its 3′-UTR	None reported	Upregulated miR-130b promoted GC cell proliferation and inhibited apoptosis by repressing *CYLD*.	[70]
	miR-20a negatively regulated *CYLD* expression by targeting its 3′ UTR	None reported	Upregulated miR-20a augmented the resistance of GC cells to cisplatin by inhibiting *CYLD* and activating NF-κB signaling and its downstream targets.	[71]
	miR-425-5p negatively regulated *CYLD* expression by targeting its 3′ UTR	None reported	Upregulated miR-425-5p may promote GC cell migration and invasion by repressing *CYLD*.	[72]
	CYLD expression was inversely correlated with hypermethylation of its promoter, which could be induced by some infectious agents	None reported	Decreased CYLD level may be associated with gender, patient’s age, high grade, and no lymph-node metastasis in GC patients.	[73]
	LncRNA CRAL functions as a ceRNA to upregulate *CYLD* expression via absorbing miR-505	None reported	LncRNA CRAL improved cisplatin resistance via the miR-505/CYLD/AKT axis in GC cells.	[74]
	miR-454 inhibited *CYLD* expression via targeting its 3′-UTR	None reported	miR-454 supported survival and induced oxaliplatin resistance in GC cells by repressing *CYLD*.	[75]
	Exosomal miR-588 from M2 polarized macrophages inhibited *CYLD* expression via targeting its 3′-UTR	None reported	Exosomal miR-588 contributed to cisplatin GC cell resistance by repressing *CYLD*.	[76]
	ZNF333 decreased CYLD level by binding to its promoter	None reported	ALKBH5 promoted the process of bile-acid-induced gastric intestinal metaplasia by ZNF333/CYLD/CDX2 pathway.	[77]
UCHs				
UCHL1	*UCHL1* may be epigenetically inactivated via promoter methylation	None reported	*UCHL1* methylation was correlated with poor clinical outcome in GC patients.	[78,79,80,81,82]
	None reported	None reported	UCHL1 promoted GC cell proliferation and metastasis by activating the Akt and ERK1/2 pathways.	[83]
	None reported	None reported	*UCHLI* silencing inhibited GC cell proliferation and metastasis.	[84]
UCHL3	None reported	None reported	UCHL3 stimulated GC metastasis by upregulating IGF2.	[85]
UCHL5	None reported	None reported	A certain subgroup of GC patients with high expression of UCHL5 had better prognosis.	[86]
	None reported	NFRKB	LncRNA DRAIC inhibited GC proliferation and metastasis by mediating ubiquitination degradation of NFRKB by interfering with its combination with UCHL5.	[87]
BAP1	None reported	None reported	BAP1 downregulation predicts unfavorable survival in GC.	[88]
OTUs				
OTUB1	None reported	None reported	OTUB1 enhanced tumor invasiveness and predicted a poor prognosis in GC.	[89]
OTUB2	None reported	KRT80	OTUB2 enhanced KRT80 stability via deubiquitination and promoted GC proliferation.	[90]
	None reported	KDM1A	OTUB2 promoted GC tumorigenesis by enhancing KDM1A-mediated stem cell-like properties.	[91]
A20	A20 expression was inversely correlated with methylation at specific CpG sites in its intronic region	None reported	Increased A20 levels were associated with poor clinical outcomes.	[92]
	None reported	None reported	*A20* depletion inhibited the capacity of proliferation, migration, and invasion of GC cells.	[93]
	miR-200a inhibited *A20* expression by targeting its 3′-UTR	RIP1	miR-200a prevented RIP1 polyubiquitination and enhanced TRAIL sensitivity by targeting *A20* in GC cells.	[94]
	*H. pylori*-induced NF-κB activation elevated *A20* expression	Procaspase-8	Increased *A20* expression by USP48 inhibited K63-linked ubiquitinylation of procaspase-8, restricting caspase-8 activity and apoptosis in GC cells.	[95,96,97,98]
	*H. pylori* infection decreased *A20* expression by upregulating miR-29a-3p	None reported	*A20* silencing promoted gastric epithelial cell migration.	[99]
JAMMs				
PSMD14	None reported	PTBP1	PSMD14 stabilized PTBP1 to promote GC cell proliferation and invasion.	[100]
CSN5	None reported	RUNX3	CSN5 facilitates nuclear export and degradation of RUNX3.	[101]
	None reported	None reported	*CSN5* knockdown inhibited proliferation and promoted apoptosis of GC cells by regulating p53-related apoptotic pathways.	[102]
	None reported	p14ARF	CSN5 potentiated GC progression by decreasing p14ARF expression through non-ubiquitin pathway.	[103]
	The activity of CSN5 was enhanced by IKKβ, and IKKβ expression was inhibited by DAPK1	PD-L1	Overexpression of DAPK1 enhanced NK cell killing and suppressed tumor immune evasion in GC cells by inhibiting the IKKβ/CSN5/PD-L1 axis.	[104]
BRCC3	LncRNA TMPO-AS1 functions as a ceRNA to upregulate *BRCC3* expression via absorbing miR-126-5p	None reported	TMPO-AS1 accelerated GC cell proliferation, migration, and angiogenesis by regulating miR-126-5p/BRCC3 axis and activating PI3K/Akt/mTOR signaling.	[105]
MJDs				
Ataxin-3	None reported	None reported	Decreased Ataxin-3 expression correlated with clinicopathologic features of GC.	[106]

**Table 2 cancers-14-05831-t002:** Compounds that suppress GC by modulating DUBs.

Compound	Structure	Affected DUB	IC_50_ (μM)	Effects	Ref.
Emetine	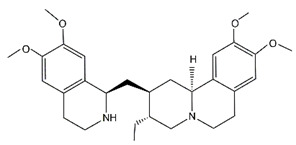	USP2	Not applicable	Emetine inhibited autophagy and GC progression by blocking USP2-E2F4 interaction	[19]
Almac4	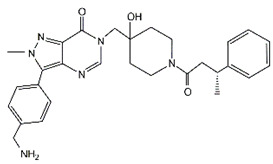	USP7	0.0015 ± 0.001	Almac4 treatment enhanced the sensitivity of GC cells to T-cell killing and inhibited GC cell proliferation by elevating p53	[26,107]
C9	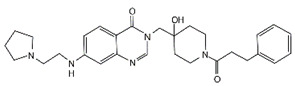	USP7	4.86	C9 suppressed proliferation of MGC-803 GC cells by decreasing MDM2 expression and thus increasing p53 and p21 levels	[108]
IU1	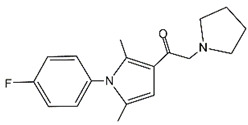	USP14	4.7 ± 0.7	IU1 restrained cell growth and tumor-promoting effects induced by YTHDF1 in GC cells	[39,109]
Compound 19	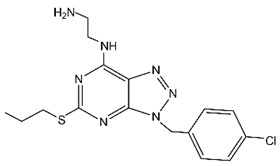	USP28	1.10 ± 0.02	Compound 19 bound to USP28 and inhibited malignant behaviors of GC cells by downregulating LSD1 and c-Myc	[110]
Lanatoside C	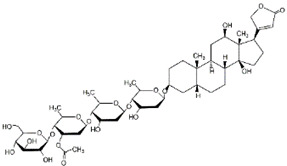	USP28	Not applicable	Lanatoside C suppressed cell proliferation and induced apoptosis by partially attenuating the binding between USP28 and c-Myc in GC cells	[111]
Galangin	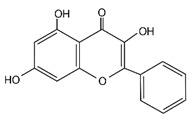	UCHL1	Not applicable	Galangin-induced growth inhibitory effect in SNU-484 GC cells was accompanied by UCHL1 upregulation	[112]
3,3′-Diindolylmethane	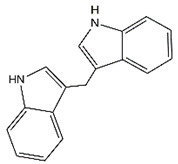	BAP1	Not applicable	3,3′-Diindolylmethane induced ferroptosis in BGC-823 GC cells by BAP1 upregulation	[113]

IC_50_, the half maximal inhibitory concentration.

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
