# Peer review of "Research Progress for Targeting Deubiquitinases in Gastric Cancers"

_cancers, 2022, doi:10.3390/cancers14235831_

Round 1

Reviewer 1 Report

This review aims to summarize the role of DUBs in gastric cancers.  It is informative. Some precisions on the molecular functions or role of some targets of the DUBs should improve the understanding of the article (expanded below).

Could authors precise the molecular function of PRC2 complex and its role in promoting invasion and migration (Line 94), the function of DDX54 (line 104), ALOX15 (line 135), EZH2 (line 180), m6A and FDZ7 (line 212-219), BMI& (line 241), BTBP1 (line 438), the role of the demethylase KDM1A and keratin KRT80 in GC (line 407-408), the role of RIP1 in GC (line 422).

Line 127: the authors indicated that “this study provided … RNA metabolism”. I did not understand which study are they talking about? It seems to be the reference #34 indicating that USP11 can affect RhoA and Ras. In that case, I do not understand the relationship between RhoA and RAS-mediated signaling pathways and RNA metabolism?  

I think that the part on the antitumoral activities of DUB inhibitors deserves to be developed a little more.  

Author Response

Dear reviewer professor,

On behalf of my co-authors, we sincerely appreciate your positive comments and suggestions on our manuscript entitled as " Research progress for targeting deubiquitinases in gastric cancers " (ID number: 2018923). We have revised the manuscript according to your suggestions and comments. All changes were marked up using the “Track Changes” function in the revised manuscript. We hope that you will now find it suitable for publication in the Cancers. Our point to point addressing to the comments are detailed on the attachment.

We are looking forward to hearing from you at your earliest convenience.

Yours sincerely

Jingjing Hou

Response to the Reviewer 1’s comments:

This review aims to summarize the role of DUBs in gastric cancers. It is informative. Some precisions on the molecular functions or role of some targets of the DUBs should improve the understanding of the article (expanded below).

Could authors precise the molecular function of PRC2 complex and its role in promoting invasion and migration (Line 94), the function of DDX54 (line 104), ALOX15 (line 135), EZH2 (line 180), m6A and FDZ7 (line 212-219), BMI& (line 241), BTBP1 (line 438), the role of the demethylase KDM1A and keratin KRT80 in GC (line 407-408), the role of RIP1 in GC (line 422).

Response: Thanks for your positive evaluation. The molecular functions of targets you suggested have been delineated in the revised manuscript (line 98-103, 111, 143-144, 190, 224, 227, 255, 459, 425-427, 442-444).

Line 127: the authors indicated that “this study provided … RNA metabolism”. I did not understand which study are they talking about? It seems to be the reference #34 indicating that USP11 can affect RhoA and Ras. In that case, I do not understand the relationship between RhoA and RAS-mediated signaling pathways and RNA metabolism? 

Response: Thanks for your concern. “This study” in the sentence “this study provided … RNA metabolism” refers to the reference #25, which indicates that DHX40 and DDX24, two DEAD/DEAH-box RNA helicases, as novel targets of USP7. We have rewritten this part to avoid this misleading in the revised manuscript (Page 10, Line 130-137). In addition, we have paid tremendous attention to revise grammatical and spelling mistakes during revision, and the manuscript has been revised for language polishing by Editage editing service (https://www.editage.cn/). We hope it is now suitable for publication.

I think that the part on the antitumoral activities of DUB inhibitors deserves to be developed a little more.

Response: Thanks for your advice. Since the antitumor activities of DUB inhibitors has been extensively reviewed, we just cited the related literatures to introduce this point in the previous manuscript. As you suggested, we further illustrate the antitumor activities of DUB inhibitors by enumerating the application of several DUB inhibitors in different cancer types including squamous cell carcinoma, breast cancer and colon cancer (Page 18, Line 510-513).

Again, we appreciate all of your insightful comments. Thank you for taking the time and energy to help us improve the paper.

Reviewer 2 Report

Tao An and colleagues have written a nice review summarizing the role of DUBs in gastric cancer. The review is nicely written and upon addition of some minor changes, I can recommend the publication of the article in Cancers. 

Minor comments:  - For USP28, it will be interesting to indicate that this DUB can stabilize relevant oncoproteins for gastric cancer such as JUN, NOTCH1 or CCNE (check and cite table 2 from this review: https://www.mdpi.com/2073-4409/10/10/2652/htm).    - It will be relevant to discuss the successful in vivo application of several DUB inhibitors in other cancer types such as lung Squamous tumors (please cite: https://www.embopress.org/doi/full/10.15252/emmm.201911101), Breast cancer (please cite: https://www.nature.com/articles/s41388-018-0590-8) or Colon cancer (please cite: https://www.ncbi.nlm.nih.gov/pmc/articles/PMC6339463/)    

Author Response

Dear reviewer professor,

On behalf of my co-authors, we sincerely appreciate your positive comments and suggestions on our manuscript entitled as " Research progress for targeting deubiquitinases in gastric cancers " (ID number: 2018923). We have revised the manuscript according to your suggestions and comments. All changes were marked up using the “Track Changes” function in the revised manuscript. We hope that you will now find it suitable for publication in the Cancers. Our point to point addressing to the comments are detailed on the attachment.

We are looking forward to hearing from you at your earliest convenience.

Yours sincerely

Jingjing Hou

Response to the Reviewer 2’s comments:

Tao An and colleagues have written a nice review summarizing the role of DUBs in gastric cancer. The review is nicely written and upon addition of some minor changes, I can recommend the publication of the article in Cancers.

Response: Thanks for your positive evaluation.

Minor comments:  - For USP28, it will be interesting to indicate that this DUB can stabilize relevant oncoproteins for gastric cancer such as JUN, NOTCH1 or CCNE (check and cite table 2 from this review: https://www.mdpi.com/2073-4409/10/10/2652/htm).   

Response: Thanks for your constructive comment. The literature and relevant substrates of USP28 you suggested have been added in the revised manuscript (Page 13, Line 285-286).

- It will be relevant to discuss the successful in vivo application of several DUB inhibitors in other cancer types such as lung Squamous tumors (please cite: https://www.embopress.org/doi/full/10.15252/emmm.201911101), Breast cancer (please cite: https://www.nature.com/articles/s41388-018-0590-8) or Colon cancer (please cite: https://www.ncbi.nlm.nih.gov/pmc/articles/PMC6339463/) .

Response: Many thanks to your kind reminder. The in vivo application of several DUB inhibitors in other cancer types you suggested has been included in the revised manuscript (Page 18, Line 510-513).

Again, we appreciate all of your insightful comments. Thank you for taking the time and energy to help us improve the paper.
